# Pressure Induced Stability Enhancement of Cubic Nanostructured CeO_2_
[note 1]

**DOI:** 10.3390/nano10040650

**Published:** 2020-03-31

**Authors:** Mariano Andrés Paulin, Gaston Garbarino, Ana Gabriela Leyva, Mohamed Mezouar, Joaquin Sacanell

**Affiliations:** 1Laboratorio Argentino de Haces de Neutrones, Centro Atómico Bariloche, CNEA, Av. E. Bustillo 9500, San Carlos de Bariloche, Río Negro R8402AGP, Argentina; 2European Synchrotron Radiation Facility, 71 Av. des Martyrs, Grenoble 38000, France; 3Departamento de Física de la Materia Condensada, Centro Atómico Constituyentes, CNEA, Av. Gral. Paz 1499, San Martín, Buenos Aires 1650, Argentina; 4Escuela de Ciencia y Tecnología, Universidad Nacional de General San Martín, Alem 3901, San Martín, Buenos Aires 1650, Argentina; 5Instituto de Nanociencia y Nanotecnología, CNEA-CONICET, Av. Gral. Paz 1499, San Martín, Buenos Aires 1650, Argentina

**Keywords:** nanoparticles, ceria, high pressure, X-ray diffraction, stacking faults

## Abstract

Ceria (CeO_2_)-based materials are widely used in applications such as catalysis, fuel cells and oxygen sensors. Its cubic fluorite structure with a cell parameter similar to that of silicon makes it a candidate for implementation in electronic devices. This structure is stable in a wide temperature and pressure range, with a reported structural phase transition to an orthorhombic phase. In this work, we study the structure of CeO_2_ under hydrostatic pressures up to 110 GPa simultaneously for the nanometer- and micrometer-sized powders as well as for a single crystal, using He as the pressure-transmitting medium. The first-order transition is clearly present for the micrometer-sized and single-crystal samples, while, for the nanometer grain size powder, it is suppressed up to at least 110 GPa. We show that the stacking fault density increases by two orders of magnitude in the studied pressure range and could act as an internal constraint, avoiding the nucleation of the high-pressure phase.

## 1. Introduction

Cerium dioxide, or ceria, is widely known and studied because of its uses in catalysis, as a solid oxide fuel cell material and, much more recently, as a potential component of spintronic devices due to its large dielectric constant and the appearance of magnetism in several conditions [1,2,3]. In ambient pressure and temperature conditions, ceria presents a cubic fluorite-type structure (Fm3m space group) with the Ce and O atoms occupying the high-symmetry Wyckoff positions 4a (0,0,0) and 8c (¼,¼,¼), respectively. Several works have been dedicated to the study of the stability of the crystal structure under pressure [4,5,6,7]. Above 30 GPa, a first-order phase transition to an orthorhombic structure has been shown to occur with a volume jump of 7%. However, there is no consensus regarding the space group of this phase, and both Pnam [6] and Pbnm [5] have been proposed. Micrometer-sized grains of ceria have been thoroughly studied and the equation of a state below 130 GPa [7] has been reported, using neon as the pressure-transmitting media (PTM). In addition, the influence of the particle size (powder grain size of 12 nm) was studied by Q. Wang et al. [8], where an unusual behavior below 30 GPa was observed, along with a shift in the phase transition up to at least 40 GPa. This unusual behavior, which includes a negative compressibility, was observed using silicone oil, methanol/ethanol, or without a PTM. This result was explained in terms of the difficulty in observing the high-pressure phase due to the peak broadening related to the nanometric grain size of the powder. Z. Wang et al. [9] performed a study of the pressure effect on 3 nm ceria nanoparticles up to 65 GPa, without using a PTM. In that work, no evidence of the structural phase transition was observed up to the highest applied pressure. Finally, there was a report showing a reduction in the structural phase transition to 20 GPa observed on the nanoparticles of 9–15 nm [10]. To the best of our knowledge, those are the studies performed on ceria nanoparticles for pressures above 30 GPa. In all of the mentioned cases involving nanoparticles, the use of silicon oil or a 4:1 methanol/ethanol mixture as the PTM, or their absences, does not assure the best possible hydrostatic experimental conditions. 

These discrepancies are also present in many other materials where the pressure and temperature phase diagrams of the nanomaterials differ from the bulk parents [11]. The understanding of these differences is very important as it can offer novel approaches for the engineering of original functional nanomaterials [11].

In this work, we study the crystal structure of CeO_2_ under hydrostatic pressures up to 110 GPa simultaneously for nanometer- and micrometer-sized powders as well as for a single-crystal, using He as the pressure-transmitting medium. The first-order transition is clearly present for the micrometer-sized and single-crystal samples and is consistent with the literature-reported values. While for the nanometer grain size powder, it is suppressed up to at least 110 GPa, a pressure four times larger than the one observed in the other two samples. We were extremely careful about the hydrostatic conditions during the experiments [12,13]. We carefully prepared the samples to avoid bridging between the diamonds at the highest applied pressure. We loaded two samples simultaneously in the same DAC (Diamond Anvil Cell) to guarantee the same experimental conditions. Finally, we show that the stacking fault density increases two orders of magnitude in the studied pressure range and could act as an internal constraint, avoiding the nucleation of the high-pressure phase.

## 2. Materials and Methods

We studied three CeO_2_ samples, a single-crystal (CRYSTAL, about a 10 μm thick sample), a micrometer grain size powder (MICRO) [14], and a 4 nm (NANO) grain size powder (see Appendix A), by means of high-pressure, high-resolution angular dispersive X-ray diffraction experiments (*λ* = 0.3738 Å) at ID27 beamline of the ESRF. A pair of KB mirrors allowed us to focus the X-ray beam down to 3 × 3 μm^2^. Three different experiments were performed using membrane-driven diamond anvil cells (mDAC) with the diamond culets of 300 μm and 150 μm beveled 300 μm. In all of them, ruby and copper powders were used as the pressure markers. In the case of the powder samples, the 2D XRD images obtained using a Mar CCD 165 detector were integrated using the PyFAI software [15], as implemented in the DIOPTAS [16] suite. The refinements of the lattice parameters and peak profiles were done using the Fullprof [17] and GSAS packages [18]. For the single-crystal measurements, experimental intensities were recorded also using a Mar CCD 165 detector and were reduced with the CrysAlisPro package [19]. During the data collection, the mDAC was oscillated with 1° steps in a 64° 2*θ* range.

## 3. Results

In the first and second experiments, two grains of the NANO and MICRO samples were loaded simultaneously in the same mDAC and the pressure was increased up to 70 and 110 GPa, respectively. In Figure 1a,b, it can be seen that below 26 GPa, both samples present a typical cubic fluorite structure (low-pressure phase, LP). The MICRO ceria presents narrower peaks than the NANO sample due to its larger crystallite size. Above 26 GPa, new weak reflections are observed for the MICRO sample that can be indexed with the reported orthorhombic phase (high-pressure phase, HP). The observed phase transition shows a very large coexistence region, which extends up to 70 GPa. However, we only report the unit cell volume of the LP phase up to around 60 GPa due to the low intensity of the LP phase peaks at a higher pressure. No significant changes can be observed for the NANO ceria apart from a broadening in the peaks up to the highest applied pressure of 110 GPa, extending the stability field of the LP phase to above 1 Mbar, i.e., more than four times the critical pressure observed for the MICRO ceria.

Finally, a third experiment was performed with the CRYSTAL sample. In Figure 1c,d, reconstructions of the reciprocal space layers at 10 GPa are presented showing the good crystallographic quality of the sample. Upon compressions above 30(3) GPa, the structural phase transition from the LP to the HP phase was also observed with a very small coexistence region. A severe degradation in the crystallographic sample quality, due to the nature of the transition and the associated volume reduction, was observed, avoiding any possible single-crystal analysis for the HP phase. 

In Figure 2, the pressure dependence of the LP phase lattice parameter (*a_CUBIC_*) is presented for the three samples. 

It can be clearly observed that at room pressure the MICRO and CRYSTAL samples present smaller lattice parameters than the NANO counterpart. This effect can be related to the presence of Ce^3+^, with a larger ionic radius than Ce^4+^, in the surface of the nanoparticles [20,21]. Under this assumption [22,23], we were able to estimate that an amount of 1.8% of Ce^3+^ was responsible for the observed differences in the cell parameters of the NANO sample compared with the MICRO and CRYSTAL samples (see Appendix A). This is in good agreement with the results published in the literature data, where a vacancy content of 1.7% was reported for a 3 nm Ceria [9].

For the CRYSTAL and MICRO samples, and the pressure marker (copper), the lattice parameters at each pressure were obtained by a LeBail refinement of the integrated 2D images, whereas for the NANO sample, each peak was fitted independently using a pseudo-Voigt profile in a MATLAB code, and we obtained the measured lattice parameters, $a_{m}\left(hkl\right)$, obtained for the Bragg peak (*hkl*). This was done because the LeBail refinement of the diffractogram of the NANO sample never gave a good factor of merit. 

The $a_{m}\left(hkl\right)$ differences increase for pressures above ~50 GPa, see Figure 2a. In Figure 2b, we show the logarithmic derivative of the lattice parameters respective to the pressure, $\chi_{hkl}$, for both the MICRO and NANO samples. We can also clearly observe a deviation in $\chi_{111}$ and $\chi_{200}$ above 50 GPa, as observed in the ratio $\chi_{200}/\chi_{111}$ plotted in Figure 2c.

## 4. Discussion

In order to clarify the origin of this discrepancy, we considered the possible effect of non-hydrostatic conditions, as proposed by A.K. Singh [24]. As is discussed in the Appendix A, non-hydrostatic conditions could be excluded as the origin of the variation in the $a_{m}\left(hkl\right)$. 

As already mentioned, cerium dioxide presents a cubic fluorite with the Ce^4+^ ions forming a face center cubic (fcc) structure, and with most fcc compounds, stacking faults along the crystallographic (111) direction are very susceptible to occur [25,26]. Using the formalism developed by *E*. Warren [26,27], the effect of stacking faults on fcc crystals can be expressed by the shifts of the peak position and its broadening. Many examples studying stacking fault density have already been published in fcc nanomaterials [28]. The peak shift in the (*hkl*) reflection, $∆{\left(2\theta \right)}_{\left(hkl\right)}^{{}^{\circ}}$, can be calculated in terms of the stacking fault probability *α*, and in particular for the (111) and (200) peaks, it can be expressed as follows
(1)$$∆{\left(2\theta \right)}_{\left(111\right)}^{{}^{\circ}}=\left(1/4\right)\;90\;\sqrt{3}\;\alpha \tan\left(\theta_{\left(111\right)}\right)/\pi^{2}$$
(2)$$∆{\left(2\theta \right)}_{\left(200\right)}^{{}^{\circ}}=\left(-1/2\right)\;90\;\sqrt{3}\;\alpha \tan\left(\theta_{\left(200\right)}\right)/\pi^{2}$$
where $\;\theta_{\left(111\right)}$ and $\;\theta_{\left(200\right)}$ are the experimental-obtained positions of the (111) and (200) reflections, respectively. The effect of stacking faults can be also calculated on the higher-order reflections, (220) and (311) replacing the prefactor (1/4) and (−1/2) by (1/4) and (−1/11), respectively.

Using Equation (1) and Equation (2), we can calculate and define $a_{\mathrm{NANO}}$ as the lattice parameter without a stacking fault effect as $a_{\mathrm{NANO}}=a_{\left(hkl\right)}+∆a_{hkl}$. In that case, we can express:(3)$$a_{\mathrm{NANO}}=a_{\left(111\right)}+∆a_{\left(111\right)}=a_{\left(200\right)}+∆a_{\left(200\right)}$$

Using the Bragg law and some algebra, we can express $\Delta a_{hkl}$ in terms of $\Delta {\left(2\theta \right)}_{\left(hkl\right)}$, and finally α in terms of $\theta_{\left(111\right)}$ and $\theta_{\left(200\right)}$, as the following:(4)α=(16 π/3)[sin(θ(111))−3/2 sin(θ(200))][2 sin(θ(111))+3/2 sin(θ(200))]

Using Equation (4), we obtained the pressure dependence of α, as can be seen in Figure 3. We can observe that at room pressure it shows a value of 0.02 that increases almost linearly to 0.04 up to 20 GPa, and then remains constant up to around 50 GPa, the pressure at which another linear increase is observed up to the highest applied pressure. This graph shows that the density of stacking faults increases dramatically above 50 GPa, reaching 0.3 at 110 GPa, more than two orders of magnitude higher than its value at room pressure.

Consequently, we could correlate the enhancement of the stability field of the room pressure cubic structure up to at least 110 GPa in the NANO sample with the withdrawal of the nucleation centers of the high-pressure phase, due to the increase in the stacking fault density above the critical pressure observed in the MICRO and CRYSTAL samples.

Similar behavior was observed in artificially synthesized fcc Ru nanoparticles with diameters ranging from 2.4 to 5.4 nm, in which, even though the nanoparticle size increased, the grain growth did not occur due to the high stacking fault densities up to 0.48 ± 0.20 [28]. It was also reported that the retention of high-temperature structural phases at room temperature in the ZrO_2_ nanostructured compounds occurred [29,30,31]. In our case, the increase in the stacking fault density and the reduced crystallite size could act as an internal constraint that inhibits the nucleation of the high-pressure orthorhombic phase, extending the stability field of the cubic low-pressure phase. In order to gain a deeper thermodynamic analysis, the contributions of both the stacking fault density and nanostructuration effects should be considered in the total free energy of the cubic and orthorhombic phases. These extra contributions could increase the relative enthalpy difference of the cubic phase respective to the orthorhombic phase, inducing the stabilization of the cubic phase at pressures well above the critical pressure observed in the MICRO and CRYSTAL samples. 

Once α is obtained, we are able to calculate $\alpha_{\mathrm{NANO}}$, using:(5)$$\alpha_{\mathrm{NANO}}=\lambda \;\sqrt{h^{2}+k^{2}+l^{2}}/\left(2\sin\left(\theta_{\left(hkl\right)}+\Delta \theta_{\left(hkl\right)}\right)\right)$$

In Figure 4, we plot the experimental pressure dependence of the unit cell volume obtained for the CRYSTAL, MICRO and NANO samples. We can observe a very good agreement of the data obtained for the NANO sample with the (111) and (200) reflections, but also for the independent (220) and (311) reflections. By using a third-order Birch–Murnaghan equation of state (EOS), as shown in Equation (6), we got the room pressure volume (*V*_0_), the bulk modulus (*K*_0_) and its first pressure derivative (*K*^’^_0_).
(6)$$P=3/2K_{0}\left[{\left(V_{0}/V\right)}^{7/3}-{\left(V_{0}/V\right)}^{5/3}\right]\left\{1+3/4\left(K_{0}^{\prime }-4\right)\left[{\left(V_{0}/V\right)}^{2/3}-1\right]\right\}$$

A very good agreement between the experimental data and the fit is obtained, as shown in the insert of Figure 4, where the difference between the data and fit is plotted in the function of pressure. For comparison, the difference is plotted for the MICRO and NANO samples obtaining deviations below 1 GPa, even at the maximum applied pressure. The obtained parameters for the EOS are shown in Table 1 for the three samples. We obtained for the MICRO and CRYSTAL samples a bulk modulus (*K*_0_) value in very good agreement with the reported values. In the case of the NANO sample, the *K*_0_ is significantly smaller.

## 5. Conclusions

In conclusion, we studied the crystal structure of CeO_2_ under hydrostatic pressures up to 110 GPa simultaneously for samples with crystallite sizes covering four orders of magnitude, using He as the pressure-transmitting medium. A first-order phase transition occurring at 26(1) and 30(3) GPa is present for the MICRO and CRYSTAL samples, respectively, while for the NANO sample, it is inhibited up to a pressure at least four times larger. We showed that the stacking fault density increases by two orders of magnitude in this pressure range and that it could act as an internal constraint, avoiding the nucleation of the high-pressure phase. 

Further thermodynamic analysis should consider the contributions of both the stacking fault density and nanostructuration effects in the total free energy of the cubic and orthorhombic phases. A detailed analysis of these extra contributions could increase the relative enthalpy difference of the cubic phase respective to the orthorhombic phase, inducing the stabilization of the cubic phase at pressures well above the critical pressure reported for the MICRO and CRYSTAL samples. A dedicated work is in progress to evaluate these contributions. Finally, the control of phase transitions through the manipulation of stacking faults can benefit particular applications as it opens a path for the study of other compounds in which the retention of a particular phase could be affected via nanostructuration and pressure or doping, as has been shown for several ZrO_2_ compounds. 

## Figures and Tables

**Figure 1 nanomaterials-10-00650-f001:**
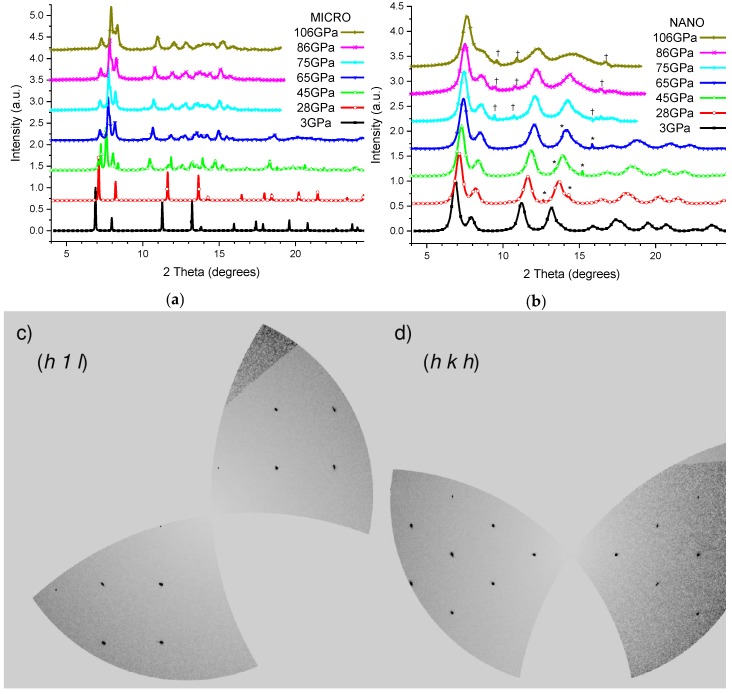
X-ray diffraction patterns for different pressure of MICRO (**a**) and NANO (**b**) CeO_2_, symbols * and † represent reflections from Helium and Rhenium, respectively. (**c**) and (**d**) show reciprocal space reconstructions of (*h1l*) and (*hkh*) layers, respectively, of the single-crystal at 10 GPa.

**Figure 2 nanomaterials-10-00650-f002:**
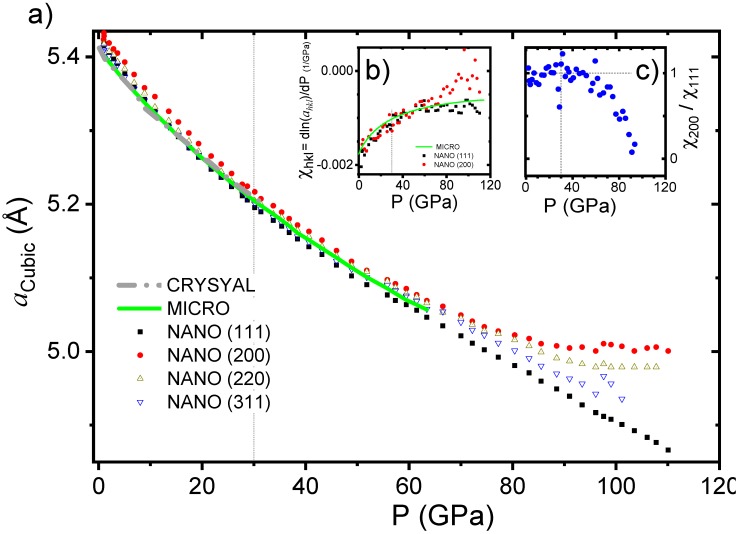
(**a**) Lattice parameters of the cubic structure (*a*_CUBIC_) for the CRYSTAL, MICRO and NANO samples in the function of pressure. In the case of the NANO sample, the cubic lattice parameter was calculated for each independent reflection ((111), (200), (220) and (311)). (**b**) Pressure dependence of the logarithmic derivative of the lattice parameter respect to the pressure, $\chi_{hkl}$ for both the MICRO and NANO samples. (**c**) Pressure dependence of the ratio $\chi_{200}/\chi_{111}$; a clear deviation from pressure above 50 GPa can be observed.

**Figure 3 nanomaterials-10-00650-f003:**
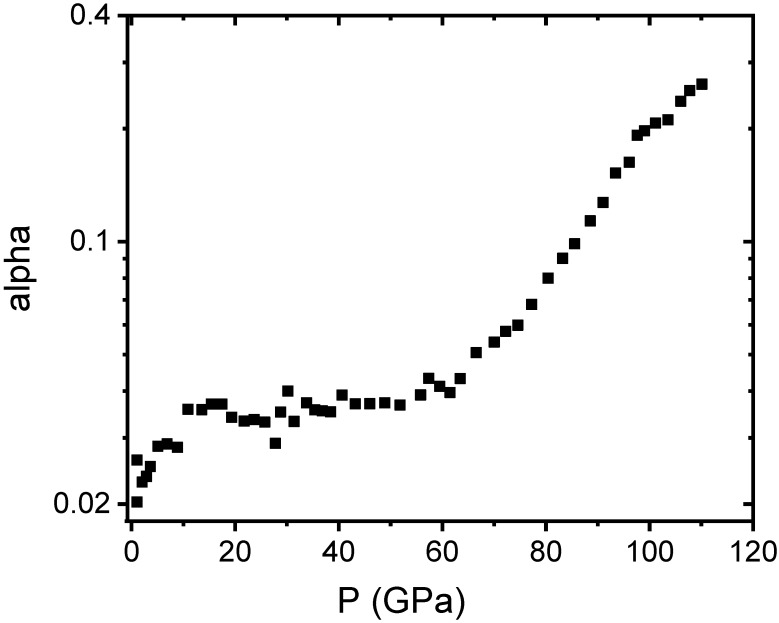
Pressure dependence of the stacking fault density calculated for the NANO sample using Equation (4), see text.

**Figure 4 nanomaterials-10-00650-f004:**
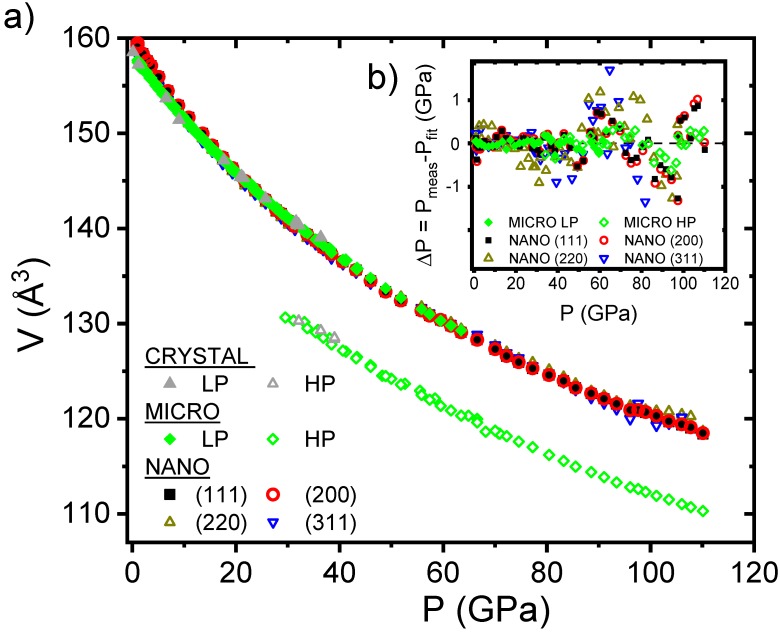
(**a**) Pressure dependence of the unit cell volume obtained using a LeBail refinement for the CRYSTAL and MICRO samples and using Equation (5) for the NANO sample in order to get the volume without the effect of stacking faults (see text). (**b**) Pressure dependence of the difference between the experimental data and the third-order Birch–Murnaghan equation of state.

**Table 1 nanomaterials-10-00650-t001:** The equation of state parameters for the CRYSTAL, MICRO and NANO samples. For the CRYSTAL sample, only the low-pressure phase parameters are presented. For the MICRO sample, the low- and high-pressure parameters are shown.

Sample/Parameter	*V*_0_ (Å^3^)	*K*_0_ (GPa)	*K* ^′^ _0_
CRYSTAL low pressure	158.4 ± 0.2	195 ± 10	5.2 ± 0.7
MICRO low pressure	158.39 ± 0.04	202 ± 1	4.25 ± 0.04
MICRO high pressure	146.2 ± 0.4	219 ± 6	4.1 ± 0.1
NANO	159.9 ± 0.2	169.5 ± 3	5.2 ± 0.1

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
