# Peer review of "Pressure Induced Stability Enhancement of Cubic Nanostructured CeO2"

_nanomaterials, 2020, doi:10.3390/nano10040650_

Round 1

Reviewer 1 Report

Authors reported that pressure induced stability enhancement of cubic nanostructured CeO2. The sample is widely known and studied because of its uses in catalysis. Therefore, we are one of interest.

The lattice parameter, stacking fault density and unit cell volume constant is properly discussed and useful information has been reported. Therefore, this paper is worthy of acceptance.

It requires formal correction with one formal content. The references cited in the conclusion are not relevant. Should be cited in the text.

Author Response

We thank this reviewer for his/her useful comments.

Following the suggestions of the referee, we rearranged the references, changing those located in the conclusions to the main text of the article.

Reviewer 2 Report

The authors investigated the structure of CeO2 (ceria) under pressure up to 110 GPa simultaneously for bulk powder, nano-powder, and single crystal CeO2 using He as a pressure transmitting medium. CeO2 is used as basis for materials in applications such as catalysis, fuel cells and oxygen sensors. Its high-pressure behavior has been studied before, but reports were not consistent regarding the space group of the high-pressure orthorhombic phase, and the assurance of hydrostatic pressure conditions as well as the effect of crystallite size on phase formation. In this carefully controlled study, where truly hydrostatic pressure conditions were employed, the authors show that the significantly increased stacking fault density in nano-ceria acted as an internal constraint avoiding the nucleation of the first order structural phase transition. This may have caused the lack of phase transition in the nanocrystalline ceria also observed before.

I consider the described reported sufficiently detailed, the experimental work well-planned. The methodology was adequately described allowing others to reproduce the results as needed. The control of phase transitions through the manipulation of stacking faults can benefit future applications by opening a pathway for synthesis of particular structures via nano-structuring.

My only recommendation to strengthen the paper would be to argue why the observed phenomenon occurs from a thermodynamic point-of-view. The only formatting issue I found was that lines 68-73 seem to be a repetition of previous sentences just above them.

Author Response

We also thank this reviewer for his/her useful comments.

- We corrected repetition of previous sentences observed in lines 68-73.

- Regarding the thermodynamic description of the stability enhancement of cubic nanostructured CeO2, we could argue that the stacking fault density increase could act as an internal constraint that avoids the nucleation of the high-pressure phase. We believe that this is related to the fact that the stacking faults density and the nano-structuration effects should be considered in the total free energy of both the cubic and orthorhombic phases. These extra contributions increase the relative enthalpy difference of the cubic phase respect to the orthorhombic one that could induce the stabilization of the cubic phase at pressures well above the critical pressure observed in the micrometer grain size and single crystal samples.

We added a phrase to clarify this point in the main text and conclusions.
